# Task-Constrained Optimization for Entity-Relation Extraction

## Abstract

Multi-task learning for entity–relation extraction often suffers from implicit task interference and the absence of explicit mechanisms for structural task prioritization. We propose *Hyperbolic Barrier-based Adaptive Hierarchical Optimization*, a constraint-driven framework that treats entity recognition as a dynamic hard constraint via a numerically stable hyperbolic barrier, while adaptively reweighting relation classification through a curriculum-based thresholding strategy. This principled approach enforces strict task prioritization throughout training, yielding up to 6.4% absolute gains in triplet F1 across five benchmarks. Furthermore, the method generalizes effectively to structurally divergent domains such as recommender systems. These findings underscore that explicitly modeling task hierarchies through constrained optimization represents a critical yet underexplored paradigm for achieving stable and effective multi-task learning.

## 1 Introduction

Entity–relation extraction (ERE) aims to identify structured triples (subject, relation, object) from unstructured text, serving as a foundation for structured knowledge acquisition (Zhang et al., 2025a). The task is inherently multi-faceted, requiring the joint modeling of entity recognition and relation classification. Recent advances in unified ERE models (Zhang et al., 2025a; Yang et al., 2023b) have achieved notable progress via end-to-end architectures. However, a critical asymmetry remains under-addressed: *relation classification is only meaningful when entity boundaries are correctly identified*. Without reliable entity boundaries, relation classification becomes ill-defined—akin to inferring a friendship without knowing who the individuals are. This asymmetry gives rise to two central challenges. First, current models lack *structural prioritization*: they treat entity recognition and relation classification as peers, despite their hierarchical dependence. Second, shared optimization often results in *training rigidity*: relation losses, which typically converge faster, may prematurely dominate training and suppress entity learning, ultimately degrading triplet extraction.

To address these issues, we propose a new optimization paradigm that encodes the dependency structure directly into the objective, rather than heuristically balancing tasks. We introduce **H**yperbolic **B**arrier-based **A**daptive **H**ierarchical **O**ptimization (**HB-AHO**), a principled framework that enforces task priorities through differentiable constraints. HB-AHO treats entity recognition as a dynamic hard constraint and adaptively reweights relation optimization using a smooth hyperbolic barrier function together with a curriculum-guided thresholding strategy. This constraint-driven mechanism ensures that learning progresses in alignment with the task hierarchy, yielding both empirical and theoretical benefits.

In summary, our contributions are as follows:

- We propose HB-AHO, a general constrained optimization framework for multi-task learning that explicitly encodes task hierarchies via dynamic hard constraints. To enable smooth and stable optimization, we design a numerically stable hyperbolic barrier function and a curriculum-guided scheduling strategy. We further provide theoretical analysis demonstrating monotonicity, Lipschitz continuity, and feasibility preservation.

- The advantages of HB-AHO are validated on five diverse ERE benchmarks, achieving up to 6.4% absolute gains in triplet F1 over strong baselines. Beyond ERE, HB-AHO is also

demonstrated to generalize effectively to structurally distinct domains such as multi-task recommendation, underscoring its broad applicability.

## 2 RELATED WORK

### 2.1 MULTI-TASK LEARNING PERSPECTIVES ON ERE

ERE jointly predicts structured triples by identifying entities and classifying their relations. Existing models typically fall into two categories, namely, parameter-sharing methods and joint-decoding methods. Parameter-sharing methods, such as TDEER (Li et al., 2021) and TPLinker (Wang et al., 2020), employ shared encoders with separate decoders. These approaches reduce redundancy but often suffer from weak cross-task coupling and redundant optimization efforts (Duan et al., 2023; Sun et al., 2024). In contrast, joint-decoding methods, such as OneRel (Shang et al., 2022) and UNIRE (Wang et al., 2021), predict complete triples in a single step, enhancing output coherence. However, they often *neglect the asymmetric dependency* between tasks, where relation prediction critically depends on accurate identification of entity boundaries.

Recent advances have sought to enhance task interaction through hypergraph structures (Yan et al., 2023), bidirectional update mechanisms (Qian et al., 2024), feature-enhanced modules (Zhou et al., 2019; Wang et al., 2025), and text-to-graph generation (Zaratiana et al., 2024). Document-level reasoning approaches (Chen, 2025) and domain-specific designs for biomedical extraction (Liu & Qi, 2025) further demonstrate the importance of robust entity modeling. Nevertheless, these methods still optimize both tasks simultaneously without enforcing a strict learning order.

### 2.2 TASK COORDINATION AND PRIORITIZATION IN MULTI-TASK LEARNING

Multi-task learning (MTL) aims to improve generalization on all tasks by jointly optimizing multiple related tasks (Boyd & Vandenberghe, 2004; Caruana, 1997). Traditional MTL frameworks adopt parameter-sharing backbones—such as tower-based or expert-based models—to facilitate cross-task information flow (Yang et al., 2025). However, such designs often encounter gradient conflicts and negative transfer, especially when task objectives are unequally scaled or loosely related (Navon et al., 2022). To address these challenges, various techniques have been proposed. Dynamic loss weighting methods, such as GradNorm (Chen et al., 2018) and PCGrad (Yu et al., 2020), aim to balance learning rates or mitigate gradient interference. Pareto optimization methods, including IMTL-GG (Liu et al., 2021), MoE-MTL (Zhang et al., 2025b), and DRGrad (Liu et al., 2025), formulate MTL as a multi-objective optimization problem. Task prioritization strategies, such as Lagrangian-based formulations (Cheng et al., 2025) and connection-based methods (Li et al., 2025), dynamically adjust the focus among tasks.

Despite these efforts, most approaches *lack explicit architectural enforcement* of task hierarchy. Methods like gradient ranking (Mahapatra et al., 2023) or distillation-based ranking (Tang et al., 2024) encode task importance implicitly, without explicitly specifying when or to what extent subordinate tasks should be subordinated to foundational ones. In contrast, our method introduces a principled constrained optimization framework with differentiable hard constraints. By embedding task hierarchy as an inherent part of the optimization architecture—rather than as an auxiliary tuning strategy—we ensure that the learning process faithfully adheres to the structural dependencies among tasks.

## 3 HYPERBOLIC BARRIER-BASED ADAPTIVE HIERARCHICAL OPTIMIZATION

ERE inherently involves hierarchical task dependencies: relation classification is only meaningful when entity boundaries are correctly identified. For example, in *"Barack Obama was born in Honolulu"*, mislabeling *Obama* as a location renders the correct *born-in* relation unreachable, regardless of how well the relation classifier performs. Such asymmetry motivates the design of an optimization paradigm that explicitly respects this hierarchical dependency.

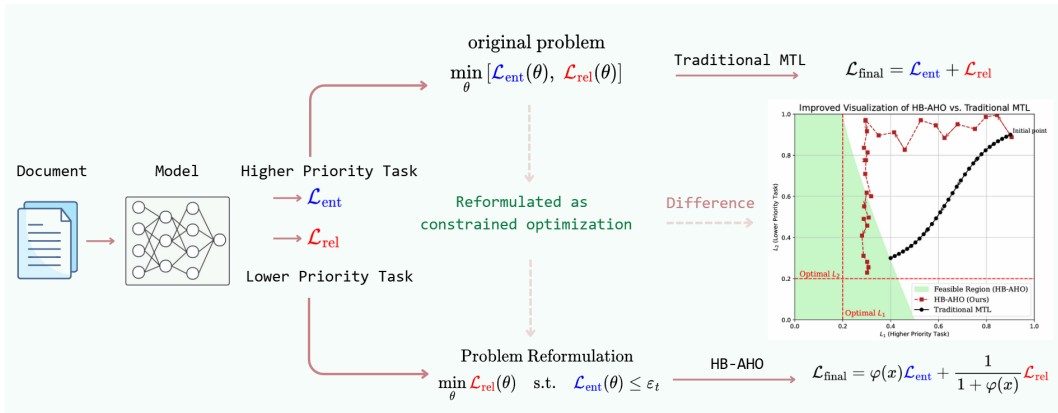

Figure 1: Conceptual overview of HB-AHO. Traditional MTL (top right) treats entity and relation losses symmetrically, leading to flat optimization $L = L_{\text{ent}} + L_{\text{rel}}$. HB-AHO reformulates the problem as constrained optimization, enforcing an entity-first regime: relation updates are gated until entity loss satisfies a dynamic threshold. The barrier-weighted loss $\varphi(x)$ ensures smooth prioritization, yielding feasible trajectories in loss space that differ fundamentally from flat MTL.

### 3.1 FROM SYMMETRIC MTL TO HIERARCHICAL OPTIMIZATION

The standard MTL formulation jointly minimizes multiple task losses:

$$\min_\theta \left[\mathcal{L}_{\text{ent}}(\theta),\ \mathcal{L}_{\text{rel}}(\theta)\right]. \tag{1}$$

This vector-valued objective problem is often approximately solved in practice using a weighted sum, which implicitly assumes that the tasks are symmetric and independently learnable. However, this assumption is violated in ERE, where relation classification is contingent upon the accuracy of upstream entity recognition. Intuitively, this is analogous to identifying the relationship between two individuals in a sentence without first recognizing who those individuals are. If entity boundaries are misidentified, relation classification becomes unreliable. Furthermore, conventional summation-based MTL suffers from *gradient imbalance*: faster-converging relation losses can prematurely dominate the optimization process, thereby suppressing the slower but foundational entity module and ultimately degrading triplet-level performance.

### 3.2 CONSTRAINED OPTIMIZATION WITH TASK HIERARCHY

To address the aforementioned limitations, we introduce HB-AHO, a constrained optimization framework that explicitly models task priority via dynamic constraints. Figure 1 contrasts traditional MTL with HB-AHO. While flat MTL optimizes entity and relation losses symmetrically, HB-AHO enforces an entity-first hierarchy through a barrier-based constraint. This reformulation yields fundamentally different optimization trajectories: relation updates are deferred until entity loss satisfies the constraint, resulting in cleaner inputs for downstream relation classification. Specifically, we enforce task prioritization by requiring the entity recognition loss to satisfy a predefined constraint before relation optimization can proceed freely. This reflects a "first-things-first" strategy: if entity boundaries remain uncertain, relation learning should be deferred—much like understanding roles in a sentence (who did what to whom) necessitates first identifying the participants. Without clearly defined entity spans, attempts to learn relations risk becoming unreliable noise. Our approach formalizes this dependency through explicit constraints rather than ad hoc heuristics. The problem is formulated as:

$$\min_\theta \mathcal{L}_{\text{rel}}(\theta) \quad \text{s.t.} \quad \mathcal{L}_{\text{ent}}(\theta) \leq \varepsilon_t. \tag{2}$$

where $\varepsilon_t$ is a dynamic threshold that governs the activation of relation training. This formulation encapsulates our core intuition: downstream optimization should be contingent upon the maturity of upstream tasks.

### 3.3 BARRIER-BASED RELAXATION AND UNIFIED LOSS

Directly solving Eq. 2 is computationally intractable in deep learning, due to the non-differentiability of inequality constraints. To circumvent this, we embed the constraint into the objective using a smooth barrier function. Classical choices (e.g., logarithmic, inverse) suffer from gradient instability near the boundary ($x \to 0^-$), which may destabilize training.

To ensure smooth optimization and bounded gradients, we propose a novel hyperbolic barrier:

$$\varphi(x) = \tanh(3x + 1), \quad \text{where} \quad x = \mathcal{L}_{\text{ent}} - \varepsilon_t. \tag{3}$$

This function is monotonic, Lipschitz continuous, and saturates smoothly, offering stable guidance near the constraint boundary.

The final composite objective becomes:

$$\mathcal{L}_{\text{final}} = \underbrace{\varphi(x)\mathcal{L}_{\text{ent}}}_{\text{Constraint Term}} + \underbrace{(1 + \varphi(x))^{-1}\mathcal{L}_{\text{rel}}}_{\text{Optimization Term}}. \tag{4}$$

This is the core of HB-AHO: a dynamic reweighting scheme that enforces task priority without breaking differentiability. When entity performance is unsatisfactory ($\mathcal{L}_{\text{ent}} > \varepsilon_t$), $\varphi(x) \to 1$ and the entity term dominates. As the constraint is gradually satisfied, relation loss regains influence. Thus, Eq. 4 encodes both **hard prioritization** (via constraint dominance) and **soft adaptability** (via weight decay).

### 3.4 ADAPTIVE THRESHOLD SCHEDULING

We further enhance flexibility by dynamically adjusting the constraint threshold. Inspired by curriculum learning, we decay $\varepsilon_t$ over time only when the constraint is satisfied:

$$\varepsilon_{t+1} = \varepsilon_t \cdot 0.95^{\delta(\mathcal{L}_{\text{ent}} \leq \varepsilon_t)}, \tag{5}$$

where $\delta(\cdot)$ is the Kronecker delta. This ensures that early training allows relaxed constraints for exploration, while later stages progressively tighten the boundary for stricter prioritization.

### 3.5 GENERALIZATION TO MULTI-TASK HIERARCHIES

While our primary focus is on ERE, HB-AHO naturally generalizes to deeper task hierarchies (e.g., detection → recognition → reasoning). For $N$ ordered tasks with losses $\{\mathcal{L}_i\}_{i=1}^N$, the loss becomes:

$$\mathcal{L}_{\text{final}} = \varphi(\mathcal{L}_1 - \varepsilon_1)\mathcal{L}_1 + \sum_{i=2}^N \frac{\mathcal{L}_i}{1 + \varphi(\mathcal{L}_{i-1} - \varepsilon_{i-1})}. \tag{6}$$

Each task is gated by the success of its predecessor, enforcing recursive priority while maintaining full differentiability. This formulation scales linearly with the number of tasks, offering both flexibility and efficiency in structured learning scenarios.

## 4 EXPERIMENTAL RESULTS AND ANALYSIS

### 4.1 EXPERIMENT SETTINGS

#### 4.1.1 EXPERIMENTAL SETUP

In ERE tasks, our model architecture follows the design illustrated in Appendix E.1. The initial threshold $\varepsilon$ in Eq. 3 is set to 0.05. All models are trained for 100 epochs with a batch size of 4. Experiments are conducted on NVIDIA RTX 4090 GPUs. Early stopping is applied based on validation triplet F1 to prevent overfitting; training is terminated if no improvement is observed within 10 consecutive epochs. The compared baseline models include: BADS (Zhou et al., 2019), SPN4RE (Sui et al., 2024), ERFD-RTE (Chen et al., 2024), ERGM (Gao et al., 2023), MFSF (Wang et al., 2025).

#### 4.1.2 DATASETS

We evaluate HB-AHO on five benchmarks spanning diverse domains, including News (NYT (Riedel et al., 2010)), Wikipedia (WebNLG (Riedel et al., 2010), DocRED (Yao et al., 2019)), and biomedicine (CDR (Li et al., 2016), GDA (Wu et al., 2019)). These datasets differ widely in relation types and instance scales, posing unique challenges for multi-task optimization. The dataset statistics are summarized in Table 1. For clarity, we categorize them into sentence-level (NYT, WebNLG) and document-level (DocRED, CDR, GDA) settings, based on whether relation reasoning is confined to a single sentence or spans multiple sentences.

Table 1: Statistics of the five ERE benchmarks used in our experiments, spanning diverse domains (news, Wikipedia, biomedicine) and task granularity. NYT and WebNLG are sentence-level datasets, whereas DocRED, CDR, and GDA require document-level reasoning. # indicates instance count per split.

| Dataset | Domain | #Train | #Val. | #Test |
|---------|--------|--------|-------|-------|
| NYT | News | 56k | 5k | 5k |
| WebNLG | Wikipedia | 35k | 1.7k | 1.7k |
| DocRED | Wikipedia | 3k | 300 | 700 |
| CDR | Biomedical | 500 | 500 | 500 |
| GDA | Biomedical | 19k | 4.7k | 4.7k |

Table 2: Results on five ERE benchmarks (NYT, WebNLG, DocRED, CDR, GDA). For each model, we report relation, entity, and triplet F1 scores **with and without** the proposed HB-AHO optimization. Rather than competing for absolute state-of-the-art, this table highlights that HB-AHO consistently improves the more challenging *entity* and *triplet* metrics across strong baselines, while relation F1 may fluctuate slightly. The results confirm HB-AHO's robustness and its effectiveness in enforcing task hierarchy, especially on document-level datasets (DocRED, CDR, GDA) where improvements in entity recognition translate into larger gains in triplet extraction.

| Model | Dataset | Without HB-AHO | | | With HB-AHO | | |
|-------|---------|----------|--------|---------|----------|--------|---------|
| | | Relation | Entity | Triplet | Relation | Entity | Triplet |
| BADS | NYT | 86.5 | 73.2 | 79.3 | 86.9 (↑0.4%) | 78.6 (↑5.4%) | 85.7 (↑6.4%) |
| SPN4RE | NYT | 92.5 | 92.2 | 92.3 | 91.7 (↓0.8%) | 94.3 (↑2.1%) | 94.8 (↑2.1%) |
| ERFD-RTE | NYT | 94.0 | 91.4 | 92.7 | 92.1 (↓1.9%) | 94.6 (↑3.2%) | 95.1 (↑2.4%) |
| ERGM | NYT | 93.3 | 91.5 | 92.4 | 93.4 (↑0.1%) | 93.6 (↑2.1%) | 94.1 (↑1.7%) |
| MFSF | NYT | 93.6 | 91.7 | 92.6 | 93.6(↑0.0%) | 94.3 (↑2.6%) | 94.8 (↑2.2%) |
| BADS | WebNLG | 85.3 | 83.1 | 84.2 | 85.7 (↑0.4%) | 86.9 (↑3.8%) | 87.6 (↑3.4%) |
| SPN4RE | WebNLG | 93.1 | 93.6 | 93.4 | 92.6 (↓0.5%) | 95.0 (↑1.4%) | 94.7 (↑1.3%) |
| ERFD-RTE | WebNLG | 91.2 | 87.4 | 89.3 | 90.7 (↓0.5%) | 92.5 (↑5.1%) | 93.6 (↑4.3%) |
| ERGM | WebNLG | 94.2 | 91.2 | 92.7 | 94.4 (↑0.2%) | 93.8 (↑2.6%) | 94.8 (↑2.1%) |
| MFSF | WebNLG | 94.9 | 92.3 | 93.5 | 94.8 (↓0.1%) | 94.7 (↑2.4%) | 95.1 (↑1.6%) |
| BADS | DocRED | 52.1 | 54.8 | 46.9 | 51.8 (↓0.3%) | 59.0 (↑4.2%) | 51.0 (↑4.1%) |
| SPN4RE | DocRED | 68.3 | 57.3 | 50.1 | 69.5 (↑1.2%) | 60.4 (↑3.1%) | 54.9 (↑4.8%) |
| ERFD-RTE | DocRED | 66.9 | 56.4 | 49.7 | 66.4 (↓0.5%) | 60.1 (↑3.7%) | 54.2 (↑4.5%) |
| ERGM | DocRED | 67.1 | 57.1 | 50.0 | 68.0 (↑0.9%) | 60.7 (↑3.6%) | 55.3 (↑5.3%) |
| MFSF | DocRED | 68.0 | 57.5 | 50.5 | 68.9 (↑0.9%) | 61.2 (↑3.7%) | 55.8 (↑5.3%) |
| BADS | CDR | 93.5 | 50.6 | 46.7 | 93.2 (↓0.3%) | 55.0 (↑4.4%) | 51.5 (↑4.8%) |
| SPN4RE | CDR | 96.6 | 52.9 | 48.3 | 96.4 (↓0.2%) | 57.8 (↑4.9%) | 53.1 (↑4.8%) |
| ERFD-RTE | CDR | 95.7 | 51.2 | 47.2 | 95.9 (↑0.2%) | 55.8 (↑4.6%) | 52.0 (↑4.8%) |
| ERGM | CDR | 96.1 | 52.1 | 47.9 | 96.2 (↑0.1%) | 56.5 (↑4.4%) | 52.6 (↑4.7%) |
| MFSF | CDR | 96.3 | 52.7 | 48.1 | 96.5 (↑0.2%) | 57.1 (↑4.4%) | 52.8 (↑4.7%) |
| BADS | GDA | 98.5 | 61.5 | 59.3 | 98.7 (↑0.2%) | 65.9 (↑4.4%) | 64.5 (↑5.2%) |
| SPN4RE | GDA | 99.1 | 62.3 | 60.3 | 99.0 (↓0.1%) | 67.5 (↑5.2%) | 65.7 (↑5.4%) |
| ERFD-RTE | GDA | 98.7 | 61.2 | 59.1 | 98.9 (↑0.2%) | 65.7 (↑4.5%) | 64.0 (↑4.9%) |
| ERGM | GDA | 99.0 | 62.0 | 60.0 | 99.2 (↑0.2%) | 66.8 (↑4.8%) | 65.3 (↑5.3%) |
| MFSF | GDA | 99.2 | 62.7 | 60.7 | 99.1 (↓0.1%) | 67.4 (↑4.7%) | 65.9 (↑5.2%) |

## 4.2 EXPERIMENTAL RESULTS

Our objective is not to pursue incremental state-of-the-art scores on already saturated datasets, but to examine whether *inserting* HB-AHO as an optimization layer consistently improves learning dynamics across architectures and data regimes. Table 2 summarizes results on five benchmarks, comparing backbones with and without HB-AHO.

On sentence-level datasets (NYT, WebNLG), where existing models are already close to ceiling, the effect of HB-AHO is moderate but systematic: entity and triplet F1 rise across nearly all backbones, while relation F1 exhibits small positive or negative fluctuations. This outcome is consistent with the design of HB-AHO, which enforces entity recognition as a prerequisite before fully engaging relation updates. In practice, this means that relation-only accuracy may occasionally decline, yet the quality of triplets—the end task—benefits from reduced boundary errors. These findings indicate that task hierarchy remains a useful inductive bias even when architectural expressiveness has largely plateaued.

On document-level datasets (DocRED, CDR, GDA), the picture is more striking. Because these tasks require reasoning over multiple sentences, error propagation from entity boundaries is amplified. Here, HB-AHO yields substantially larger improvements: entity F1 increases consistently across all backbones, and these upstream gains propagate into disproportionately higher triplet F1—often by several points. Small swings in relation F1 thus become immaterial compared to the robustness gained at the triplet level. This suggests that HB-AHO does not merely reweight losses, but actively restructures the learning trajectory to supply cleaner entity candidates for downstream relation decisions.

The credibility of these gains rests on two observations. First, improvements occur across diverse architectures (BADS, SPN4RE, ERFD-RTE, ERGM, MFSF), supporting that the effect arises from optimization rather than model-specific heuristics. Second, the empirical signature is coherent: entity F1 shows the largest gains, triplet F1 the next, while relation F1 fluctuates within a narrow band—a pattern precisely predicted by the dependency structure of the task. Taken together, the evidence demonstrates that HB-AHO functions as a general optimization layer encoding the asymmetry of ERE. Even in cases where relation F1 decreases slightly, the overall utility measured by triplet extraction improves, particularly in long-context settings where entity errors are most damaging. These results provide concrete support for the view that hierarchical optimization, rather than flat multi-task fusion, is the more principled bias for structurally dependent problems.

### 4.3 Comparisons with MTL Methods

Table 3: F1 scores of MTL optimization strategies on DocRED. HB-AHO outperforms gradient/Pareto/Lagrangian baselines in triplet F1, validating the benefit of constraint-based task hierarchy. Bold indicates best results.

| Method | Entity F1 | Relation F1 | Triplet F1 |
|--------|-----------|-------------|------------|
| SPN4RE (Baseline) | 68.3 | 57.3 | 50.1 |
| +PCGrad (Yu et al., 2020) | 68.6 (↑0.3) | 58.7 (↑1.4) | 52.6 (↑2.5) |
| +IMTL-GG (Liu et al., 2021) | 68.4 (↑0.1) | 58.2 (↑0.9) | 52.1 (↑2.0) |
| +AdaTask (Yang et al., 2023a) | 69.7 (↑1.4) | 57.6 (↑0.3) | 51.9 (↑1.8) |
| +NMT (Cheng et al., 2025) | 69.2 (↑0.9) | 58.3 (↑1.0) | 53.1 (↑3.0) |
| +DRGrad (Liu et al., 2025) | 68.1 (↓0.2) | 59.1 (↑1.8) | 52.8 (↑2.7) |
| **HB-AHO** | **69.8 (↑1.5)** | **60.4 (↑3.1)** | **54.9 (↑4.8)** |

As shown in Table 3, HB-AHO consistently outperforms widely adopted multi-task optimization baselines, including gradient-based (PCGrad), Pareto-front (IMTL-GG), and Lagrangian-based (NMT) methods. Although these approaches mitigate gradient interference through different formulations, they share a structural limitation: all assume tasks coexist symmetrically in a flat optimization space. This assumption conflicts with hierarchical problems such as ERE, where relation classification is ill-defined without reliable entity boundaries. Consequently, strategies that merely rebalance gradients or losses can reduce conflicts but cannot ensure that learning progresses in the correct order. HB-AHO addresses this misalignment by embedding task hierarchy directly into the objective via a barrier-based constraint, thereby enforcing an entity-first regime. Relation updates are naturally deferred until entity recognition reaches sufficient stability, so improvements in triplet F1 arise less from marginal relation modeling and more from preventing noisy entity predictions from propagating downstream. This distinction highlights why HB-AHO surpasses task-agnostic baselines even when their balancing strategies appear effective. More broadly, the results suggest that task prioritization in hierarchical MTL is better understood as a fundamental architectural principle rather than an

auxiliary regularizer. By treating hierarchy as a first-class constraint, HB-AHO offers a scalable, optimization-theoretic alternative that provides stronger guarantees for structurally dependent learning scenarios.

## 4.4 ABLATION STUDIES

### 4.4.1 THE EFFECTS OF DYNAMIC THRESHOLD

Table 4: Effect of initial threshold $\varepsilon$ on DocRED F1 scores. HB-AHO remains stable across a wide range of $\varepsilon$, with best performance around 0.05–0.10.

| Variant | $\varepsilon$ | Violations | Relation | Entity | Triplet |
|---|---|---|---|---|---|
| SPN4RE (Baseline) | — | — | 68.3 | 57.3 | 50.1 |
| HB-AHO | 1.00 | 34 | 69.0 (↑0.7) | 60.2 (↑2.9) | 54.1 (↑4.0) |
| | 0.50 | 30 | 69.2 (↑0.9) | 60.4 (↑3.1) | 54.4 (↑4.3) |
| | 0.10 | 15 | **69.5** (↑1.2) | 60.4 (↑3.1) | 54.3 (↑4.2) |
| | 0.05 | 7 | 69.4 (↑1.1) | **60.5** (↑3.2) | **54.5** (↑4.4) |
| | 0.01 | 1 | 69.1 (↑0.8) | 60.1 (↑2.8) | 54.2 (↑4.1) |
| | 0.005 | 0 | 68.9 (↑0.6) | 60.3 (↑3.0) | 54.2 (↑4.1) |
| | 0.001 | 0 | 69.0 (↑0.7) | 60.2 (↑2.9) | 54.1 (↑4.0) |

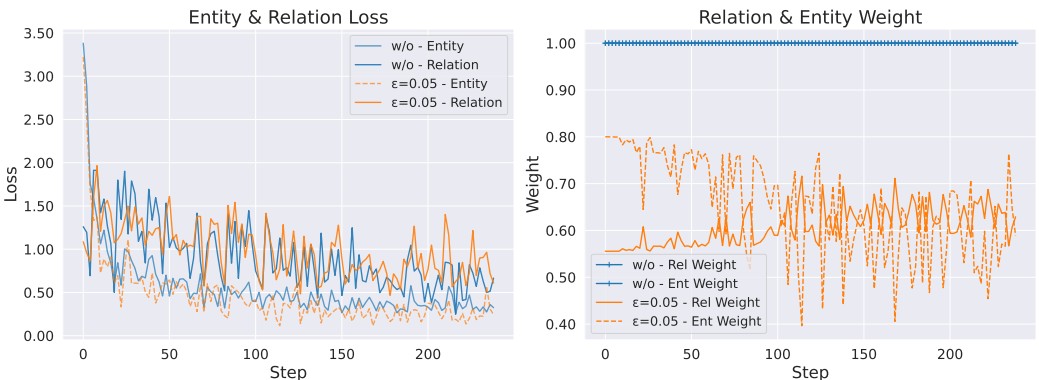

Figure 2: Training loss and dynamic task weight curves over time. HB-AHO initially prioritizes entity optimization, then gradually shifts focus to relation learning as constraints are satisfied. This behavior confirms our dynamic scheduling design for task prioritization.

To examine sensitivity to initialization, we varied the threshold $\varepsilon$ across several orders of magnitude. Figure 2 shows that, regardless of the starting value, HB-AHO always begins by emphasizing entity recognition and then gradually shifts weight to relation learning once the constraint is met; only the timing of this transition changes slightly. Table 4 confirms that the final triplet F1 remains tightly bounded (54.1–54.5, +4.0% to +4.4% over SPN4RE) across all tested $\varepsilon$. These results demonstrate that the gains of HB-AHO arise from its structural enforcement of hierarchy, not from fragile hyperparameter tuning. Larger thresholds accelerate relation updates, smaller ones delay them, yet both converge to the same balance through the barrier and curriculum mechanism. Constraint violations appear only as transient triggers for threshold decay and do not affect end performance. Taken together, the evidence shows that HB-AHO is robust to $\varepsilon$: the barrier ensures smooth weight adjustment, the curriculum harmonizes different initializations, and practitioners can simply choose any moderate value (e.g., 0.01–0.1) without hyperparameter sweeps. This robustness underscores that the improvement reflects a principled optimization design rather than parameter sensitivity.

### 4.4.2 BARRIER FUNCTION DESIGN

To assess whether HB-AHO's advantage stems from the specific barrier formulation, we compared the hyperbolic barrier against classical polynomial and neural-inspired alternatives under identical

Table 5: Ablation study of barrier function choices ($\varepsilon = 0.05$). The hyperbolic barrier provides the most balanced and robust gains in both entity and triplet F1. Red arrows = improvement; green arrows = decline.

| Type | Function | Violations | Relation | Entity | Triplet |
|------|----------|-----------|----------|--------|---------|
| SPN4RE (Baseline) | — | 7 | 68.3 | 57.3 | 50.1 |
| **Hyperbolic** | $\tanh(3x+1)$ | 8 | **69.5** (↑1.4) | **60.4** (↑3.1) | **54.9** (↑4.8) |
| Sigmoid | $(1+e^{-x})^{-1}$ | 6 | 69.2 (↑0.9) | 59.8 (↑2.5) | 54.5 (↑4.4) |
| Gaussian | $1-e^{-x^2}$ | 6 | 69.1 (↑0.8) | 59.9 (↑2.6) | 54.7 (↑4.6) |
| Softplus | $\log(1+e^x)$ | 5 | 69.0 (↑0.7) | 59.5 (↑2.2) | 54.3 (↑4.2) |
| Exponential | $e^x - 1$ | 7 | 68.9 (↑0.6) | 58.7 (↑1.4) | 53.9 (↑3.8) |
| Quadratic | $x^2$ | 10 | 57.8 (↑0.5) | 67.9 (↓0.4) | 50.0 (↓0.1) |
| Cubic | $x^3$ | 12 | 66.9 (↓1.4) | 58.6 (↑1.3) | 47.0 (↓3.1) |

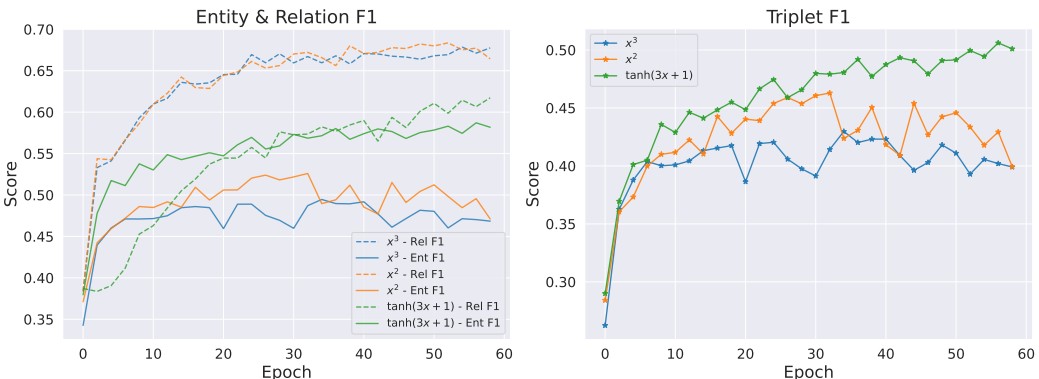

Figure 3: Training F1 trends for different barrier functions. Hyperbolic barrier enables smooth and effective optimization, while polynomial variants lead to instability, validating the robustness of our proposed design.

settings (Table 5). A consistent pattern emerges: only the hyperbolic variant simultaneously improves relation F1 (+1.4), entity F1 (+3.1), and triplet F1 (+4.8), whereas other choices either yield weaker gains or improve one component at the expense of another.

From these results we can distill three principles for effective constraint design in deep learning optimization:

- **Limitations of Polynomial Growth**: Quadratic and cubic barriers amplify small violations into unstable updates, producing severe gradient imbalance near the constraint boundary and ultimately destabilizing hierarchical training.

- **Smoothness Alone Is Insufficient**: Sigmoid and softplus functions are smooth and differentiable, yet their shallow slopes around the boundary provide inadequate responsiveness. As a result, they deliver only moderate improvements (+4.4 to +4.2 in triplet F1), underscoring that stability must be coupled with precise task-weight modulation.

- **Hyperbolic Balance**: The proposed $\varphi(x) = \tanh(3x+1)$ achieves the strongest and most balanced gains. Its strict monotonicity enforces task order consistently, its globally bounded derivative ($\|\varphi'\| \leq 3$) prevents gradient explosion, and its sigmoidal transition enables a smooth yet decisive shift from entity-dominant to relation-inclusive optimization. This geometric balance translates directly into stable training dynamics (Figure 3) and maximal triplet-level improvements.

Unlike heuristics that rely on delicate tuning, the hyperbolic barrier improves performance robustly without additional hyperparameters. This ablation not only corroborates our theoretical analysis—linking monotonicity, stability, and boundedness to effective optimization—but also highlights

that principled constraint geometry, rather than ad hoc weighting schemes, is indispensable for reliable gains in hierarchical multi-task learning.

### 4.5 CROSS-DOMAIN GENERALIZATION: BEYOND ERE TO RECOMMENDATION SYSTEMS

To assess cross-domain generalization beyond ERE, we further evaluate HB-AHO on KuaiRand1k (Yuan et al., 2022), a large-scale recommendation dataset capturing sequential user behaviors. KuaiRand1k defines eight sequential user behaviors that form a decision hierarchy, with task priority assigned from high to low as: click, long-view, like, follow, comment, forward, profile-enter, and hate. This setting differs substantially from ERE—both in modality and in the complexity of task dependencies—yet it offers an equally clear hierarchical structure.

As shown in Table 6, HB-AHO improves the average AUC of all tested backbones (STEM, Shared-Bottom, MMoE, PLE, AITM, and OMoE) by +2.1–2.6%. The strongest gains are observed on downstream, higher-commitment tasks, where accurate modeling of upstream actions provides the most leverage. These improvements are achieved without modifying the architectures themselves, confirming that HB-AHO operates as a plug-and-play optimization layer rather than an architectural tweak.

Overall, the KuaiRand1k results strengthen our central claim: HB-AHO captures structural task dependencies across domains and scales, from sentence-level extraction to multi-behavior recommendation. The fact that consistent 2.1–2.6% gains are realized on business-critical outcomes in a high-capacity industrial dataset underscores its practical impact and supports its view as a **domain-agnostic optimization principle** rather than a task-specific trick.

Table 6: Results on the KuaiRand1k dataset with eight hierarchically dependent tasks. HB-AHO consistently raises the overall *Avg. AUC* by 2.1–2.6% across diverse backbones, showing that its benefits extend beyond NLP to large-scale recommendation and to settings with more than two tasks.

| Model | Task A | Task B | Task C | Task D | Task E | Task F | Task G | Task H | Avg. AUC | MTL Gain |
|---|---|---|---|---|---|---|---|---|---|---|
| STEM | 98.7 | 98.8 | 94.9 | 90.5 | 98.8 | 91.0 | 91.7 | 98.2 | 94.6 | – |
| –HB-AHO | 99.1 | 98.5 | 95.3 | 92.6 | 99.0 | 92.0 | 92.4 | 98.0 | 96.8 | ↑**+2.2%** |
| SharedBottom | 97.9 | 99.0 | 93.6 | 88.9 | 98.3 | 88.8 | 89.2 | 92.2 | 91.8 | – |
| –HB-AHO | 98.5 | 99.1 | 94.2 | 90.1 | 98.6 | 89.9 | 90.7 | 93.1 | 94.2 | ↑**+2.4%** |
| MMoE | 98.1 | 98.9 | 94.1 | 88.6 | 98.4 | 85.6 | 90.7 | 94.1 | 93.9 | – |
| –HB-AHO | 98.7 | 99.0 | 94.6 | 90.3 | 98.7 | 87.4 | 91.5 | 95.2 | 96.0 | ↑**+2.1%** |
| PLE | 97.3 | 98.0 | 94.2 | 89.9 | 98.5 | 88.2 | 91.0 | 96.1 | 92.7 | – |
| –HB-AHO | 97.9 | 98.3 | 94.8 | 91.5 | 98.7 | 90.5 | 91.6 | 97.5 | 95.3 | ↑**+2.6%** |
| AITM | 98.6 | 98.2 | 93.7 | 89.7 | 98.1 | 88.9 | 90.0 | 97.7 | 91.7 | – |
| –HB-AHO | 99.0 | 98.1 | 94.9 | 91.0 | 98.5 | 90.2 | 91.2 | 98.4 | 94.1 | ↑**+2.4%** |
| OMoE | 97.6 | 97.9 | 94.2 | 87.8 | 98.4 | 87.0 | 90.8 | 90.9 | 92.6 | – |
| –HB-AHO | 98.2 | 98.4 | 94.7 | 89.4 | 98.7 | 89.1 | 91.2 | 92.8 | 95.0 | ↑**+2.3%** |

## 5 CONCLUSION

We introduced HB-AHO, a constraint-driven optimization framework that encodes task hierarchies through hyperbolic barrier functions. By enforcing an entity-first regime, HB-AHO improves both sentence-level and document-level ERE, with gains that are robust across diverse backbones. Beyond empirical performance, our study highlights a broader principle: hierarchical task structuring is not a matter of heuristic loss weighting, but a fundamental optimization geometry that stabilizes multi-task learning. HB-AHO demonstrates how classical constrained optimization can be reinterpreted for deep learning, yielding both theoretical guarantees and practical benefits. Looking forward, the same paradigm offers a natural foundation for domains where tasks form layered dependencies, including hierarchical reasoning, symbolic planning, and multi-modal alignment. More generally, we argue that MTL should be viewed not as a static balance among objectives, but as a guided progression across levels of abstraction—a perspective that HB-AHO makes concrete.

## REPRODUCIBILITY STATEMENT

To ensure reproducibility, we provide the following:

**Algorithm.** HB-AHO is implemented in Python/PyTorch. Source code, training scripts, and configurations are included in the supplementary material, covering barrier function, curriculum scheduling, and integration with standard ERE backbones.

**Theory.** Appendix B contains full proofs, including monotonicity, Lipschitz continuity, KKT equivalence, and stability guarantees.

**Experiments.** Section 4.2 describes the setup. Hyperparameters and training protocols are documented in Appendix E, with sensitivity analyses for $\alpha$ and $\gamma$ in Appendix B.5.

**Resources.** Experiments were run on NVIDIA RTX 4090 GPUs.

**Data.** We use five public ERE benchmarks (NYT, WebNLG, DocRED, CDR, GDA) and the KuaiRand1k dataset. Preprocessing, splits, and evaluation follow prior work.

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

## A USE OF LLMs

Large language models (LLMs) were used solely for language polishing; all technical content, methods, and experiments were developed and validated by the authors.

## B PROPERTIES OF HYPERBOLIC BARRIER FUNCTION

### B.1 DEFINITIONS AND THEORETICAL GUARANTEES

**Definition 1** (Parameterized Hyperbolic Barrier Function). *Given the constraint violation*

$$x = \mathcal{L}_{ent} - \varepsilon, \tag{7}$$

*define the barrier family as*

$$\varphi_\alpha(x) = \tanh(\alpha x + 1), \quad \alpha > 0. \tag{8}$$

*Here $\alpha$ controls the sharpness of the barrier. The choice $\alpha = 3$ is used in practice, but the theoretical analysis holds for any $\alpha > 0$.*

**Theorem 1** (Monotonicity). *$\varphi_\alpha(x)$ is strictly monotonically increasing for all $x \in \mathbb{R}$.*

**Theorem 2** (Lipschitz Continuity). *$\varphi_\alpha(x)$ is globally Lipschitz continuous with constant $L = \alpha$.*

**Theorem 3** (Feasibility Preservation and Limit Equivalence). *Consider the constrained problem*

$$\min_\theta \mathcal{L}_{rel}(\theta) \quad s.t. \quad \mathcal{L}_{ent}(\theta) \leq \varepsilon. \tag{9}$$

*For any $\alpha > 0$, if $\nabla_\theta \mathcal{L}_{final}(\theta; \alpha) = 0$ with*

$$\mathcal{L}_{final}(\theta; \alpha) = \varphi_\alpha(x)\, \mathcal{L}_{ent}(\theta) + \frac{1}{1 + \varphi_\alpha(x)}\, \mathcal{L}_{rel}(\theta), \tag{10}$$

*then there exists $\lambda^*(\alpha) \geq 0$ such that $(\theta, \lambda^*(\alpha))$ satisfies the KKT conditions of the constrained problem. Moreover, as $\alpha \to \infty$, any sequence of stationary points $\theta^*(\alpha)$ converges to a KKT stationary point of the original constrained problem.*

## B.2 PROOF OF MONOTONICITY

*Proof.* The derivative is given by

$$\varphi'_\alpha(x) = \alpha \cdot \text{sech}^2(\alpha x + 1). \tag{11}$$

Since $\text{sech}^2(z) > 0$ for all $z \in \mathbb{R}$, it follows that $\varphi'_\alpha(x) > 0$. Hence, $\varphi_\alpha(x)$ is strictly increasing. $\square$

## B.3 PROOF OF LIPSCHITZ CONTINUITY

*Proof.* $\varphi'_\alpha(x)$ is bounded by

$$|\varphi'_\alpha(x)| = \alpha \cdot \text{sech}^2(\alpha x + 1) \le \alpha. \tag{12}$$

By the Mean Value Theorem, for any $x_1, x_2 \in \mathbb{R}$:

$$|\varphi_\alpha(x_1) - \varphi_\alpha(x_2)| \le \alpha|x_1 - x_2|. \tag{13}$$

Thus, $\varphi_\alpha(x)$ is globally Lipschitz with constant $L = \alpha$. $\square$

## B.4 PROOF OF FEASIBILITY PRESERVATION AND LIMIT EQUIVALENCE

*Proof sketch.* **Step 1: KKT conditions of the constrained problem.** The Lagrangian is

$$\mathcal{L}(\theta, \lambda) = \mathcal{L}_{\text{rel}}(\theta) + \lambda(\mathcal{L}_{\text{ent}}(\theta) - \varepsilon). \tag{14}$$

Stationarity requires

$$\nabla_\theta \mathcal{L}_{\text{rel}} + \lambda \nabla_\theta \mathcal{L}_{\text{ent}} = 0, \quad \lambda \ge 0, \quad \lambda(\mathcal{L}_{\text{ent}} - \varepsilon) = 0. \tag{15}$$

**Step 2: Stationarity of the barrier formulation.** For $\mathcal{L}_{\text{final}}(\theta; \alpha)$, we have

$$\nabla_\theta \mathcal{L}_{\text{final}} = \varphi'_\alpha(x) \nabla_\theta \mathcal{L}_{\text{ent}} + \frac{1 - \varphi_\alpha(x)}{(1 + \varphi_\alpha(x))^2} \nabla_\theta \mathcal{L}_{\text{rel}}. \tag{16}$$

Setting $\nabla_\theta \mathcal{L}_{\text{final}} = 0$ gives

$$\nabla_\theta \mathcal{L}_{\text{rel}} = -\frac{(1 + \varphi_\alpha(x))^2 \varphi'_\alpha(x)}{1 - \varphi_\alpha(x)} \nabla_\theta \mathcal{L}_{\text{ent}}. \tag{17}$$

Defining

$$\lambda^*(\alpha) = \frac{(1 + \varphi_\alpha(x))^2 \varphi'_\alpha(x)}{1 - \varphi_\alpha(x)}, \tag{18}$$

we recover $\nabla_\theta \mathcal{L}_{\text{rel}} + \lambda^*(\alpha)\nabla_\theta \mathcal{L}_{\text{ent}} = 0$, with $\lambda^*(\alpha) \ge 0$. Thus $(\theta, \lambda^*(\alpha))$ satisfies the KKT conditions.

**Step 3: Limit equivalence.** As $\alpha \to \infty$:

- If $x > 0$ ($\mathcal{L}_{\text{ent}} > \varepsilon$), then $\varphi_\alpha(x) \to 1$ and $\lambda^*(\alpha) \to +\infty$, enforcing feasibility by penalizing violations.

- If $x < 0$, then $\varphi_\alpha(x) < 1$ and $\lambda^*(\alpha)$ remains finite, recovering the unconstrained relation optimization.

- If $x = 0$, then $\lambda^*(\alpha)$ is finite and positive, exactly matching the boundary case of the KKT conditions.

Hence, in the limit $\alpha \to \infty$, stationary points of $\mathcal{L}_{\text{final}}$ converge to stationary points of the constrained problem. $\square$

This result shows that the hyperbolic barrier is not only smooth and bounded (avoiding gradient explosion) but also a consistent surrogate for the original inequality constraint: it yields KKT-equivalent stationary points for finite $\alpha$, and recovers the constrained optimum in the asymptotic limit.

## B.5 Hyperparameter Sensitivity

To verify that the gains of HB-AHO are not due to fragile tuning, we examined two key hyperparameters: the parameterization of the hyperbolic barrier function $\varphi(x)$ and the curriculum decay factor $\gamma$ in Eq. equation 5. Results on the DocRED validation set are summarized below.

**Barrier parameterization.** Different hyperbolic forms yield nearly identical performance, with triplet F1 consistently improving by about $+4\%$ to $+5\%$ over the baseline. The choice $\varphi(x) = \tanh(3x + 1)$ achieves the best trade-off between gradient sensitivity and stability, but the narrow performance band across alternatives indicates that HB-AHO is robust to the exact parameterization.

Table 7: Effect of barrier parameterization on DocRED (validation set). All variants yield stable improvements; $\tanh(3x + 1)$ provides the most balanced gains.

| Form | Entity F1 ($\Delta$) | Relation F1 ($\Delta$) | Triplet F1 ($\Delta$) |
|---|---|---|---|
| $\tanh(x)$ | +4.7 | +2.2 | +4.6 |
| $\tanh(2x + 1)$ | +4.5 | +2.1 | +4.6 |
| $\tanh(3x + 1)$ | **+4.8** | **+2.3** | **+4.8** |
| $\tanh(5x + 1)$ | +4.7 | +2.2 | +4.7 |

**Curriculum decay factor.** We also varied the decay factor $\gamma \in \{0.99, 0.97, 0.95, 0.93, 0.90\}$. As shown in Table 8, triplet F1 remains within a narrow band (51.3–51.5), with $\gamma = 0.95$ providing the best balance between convergence speed and stability. Larger values slow down constraint enforcement, while smaller ones introduce mild instability, but neither substantially alters the final outcome.

Table 8: Effect of curriculum decay factor $\gamma$ on DocRED (validation set). Performance remains stable across a wide range; $\gamma = 0.95$ offers the most balanced trade-off.

| $\gamma$ | Entity F1 | Relation F1 | Triplet F1 |
|---|---|---|---|
| 0.99 | 58.7 | 64.7 | 51.4 |
| 0.97 | 58.9 | 64.6 | 51.3 |
| **0.95** | **59.0** | **64.8** | **51.5** |
| 0.93 | 58.9 | 64.8 | 51.4 |
| 0.90 | 58.9 | 64.6 | 51.3 |

**Takeaway.** Across both ablations, triplet F1 fluctuates within less than one point, demonstrating that HB-AHO's improvements stem from its structural enforcement of task hierarchy rather than from delicate hyperparameter tuning.

## B.6 Theoretical Implications and Extended Analysis

**Smoothness and Analyticity.**

**Proposition 4** (Smoothness and Analyticity). *The hyperbolic barrier function $\varphi(x)$ is infinitely differentiable over $\mathbb{R}$, i.e., $\varphi \in C^\infty(\mathbb{R})$, and is a real analytic function.*

*Proof.* The hyperbolic tangent $\tanh(z)$ is analytic on $\mathbb{R}$, and $3x + 1$ is affine. The composition of analytic functions is analytic, hence $\varphi(x)$ is analytic. $\square$

**Quasi-Convexity near the Constraint.**

**Proposition 5** (Quasi-Convexity near Constraint Boundaries). *In the neighborhood of $x = 0$, $\mathcal{L}_{final}$ is quasi-convex.*

*Proof.* Let $x = \mathcal{L}_{\text{ent}} - \varepsilon$. Near $x = 0$, expand $\varphi(x)$ via Taylor series:

$$\varphi(x) \approx \tanh(1) + 3\operatorname{sech}^2(1) \cdot x + \mathcal{O}(x^2). \tag{19}$$

Substituting into $\mathcal{L}_{\text{final}}$:

$$\mathcal{L}_{\text{final}} \approx c_1 \mathcal{L}_{\text{ent}} + c_2 \mathcal{L}_{\text{rel}} + \mathcal{O}(x^2), \tag{20}$$

where $c_1, c_2 > 0$ are constants. The Hessian of $\mathcal{L}_{\text{final}}$ is positive semi-definite in this region, implying quasi-convexity. $\square$

**Lyapunov Stability of Training Dynamics.**

**Theorem 6.** *Assume that $\mathcal{L}_{ent}$ and $\mathcal{L}_{rel}$ are smooth and bounded below. Then gradient descent on $\mathcal{L}_{final}$ is asymptotically stable and converges to a stationary point.*

*Proof.* Let $\theta(t)$ denote the training trajectory. Define the Lyapunov candidate function:

$$V(\theta) = \mathcal{L}_{\text{final}}(\theta) - \inf \mathcal{L}_{\text{final}}. \tag{21}$$

Then, along the trajectory of gradient descent:

$$\frac{dV}{dt} = -\|\nabla_\theta \mathcal{L}_{\text{final}}(\theta)\|^2 \leq 0. \tag{22}$$

This implies $V(\theta)$ is non-increasing, and by LaSalle's invariance principle, the system converges to the set of stationary points $\{\theta : \nabla_\theta \mathcal{L}_{\text{final}} = 0\}$. Hence, the training dynamics are asymptotically stable around local minima. $\square$

**Strong Duality and Slater Condition.**

**Proposition 7.** *If $\exists \theta_0$ such that $\mathcal{L}_{ent}(\theta_0) < \varepsilon$ (Slater condition), then strong duality holds.*

*Proof.* The perturbed problem

$$\min \ \mathcal{L}_{\text{rel}} \quad \text{s.t.} \quad \mathcal{L}_{\text{ent}} \leq \varepsilon + r \tag{23}$$

is convex for fixed $r$, and Slater's condition ensures zero duality gap. $\square$

**Bounded Gradient Effects.**    The derivative of the hyperbolic barrier satisfies

$$|\varphi'(x)| \leq 3, \tag{24}$$

which ensures stable gradient magnitudes throughout training, especially near the constraint boundary $x \to 0^-$. This avoids the instability common in classical barrier methods like $-\log(-x)$ or $-1/x$.

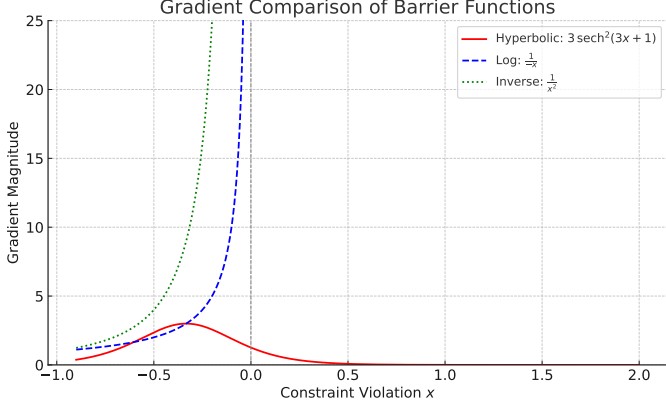

Figure 4: Comparison of $\varphi'(x)$ (hyperbolic), $-\log(-x)$, and $-1/x$ gradient behaviors near the constraint boundary.

**Task Priority Enforcement Mechanism.** The hyperbolic barrier naturally suppresses the relative influence of $\mathcal{L}_{\text{rel}}$ under severe constraint violation, by reshaping the gradient flow:

$$\frac{\partial \mathcal{L}_{\text{final}}}{\partial \mathcal{L}_{\text{rel}}} = \frac{1}{1 + \varphi(x)} - \frac{\varphi'(x)}{(1 + \varphi(x))^2} \, \mathcal{L}_{\text{rel}}. \tag{25}$$

When $\varphi(x) \to 1$, the gradient contribution from the relation loss diminishes, enforcing strict prioritization for constraint satisfaction.

**Computational Complexity.** Traditional multi-task learning approaches often rely on exhaustive grid search to identify the optimal weighting scheme for each task-specific loss. For $m$ tasks and $p$ discretized candidates per task, the total number of combinations grows as $\mathcal{O}(p^m)$.

By contrast, HB-AHO reformulates the constrained optimization problem into an unconstrained one via the hyperbolic barrier function $\varphi(x)$, enabling standard gradient-based solvers to converge in $\mathcal{O}(m)$ sequential optimization stages, corresponding to the natural hierarchical structure of task priorities. This complexity reduction not only eliminates manual tuning but also significantly improves scalability for high-dimensional multi-task learning, similar to the efficiency advantage reported in Cheng et al. (2025).

**Summary.** The hyperbolic barrier function not only empirically enhances training stability but also satisfies theoretical properties including smoothness, gradient boundedness, quasi-convexity near feasible regions, and strong duality-based feasibility guarantees. These theoretical insights establish the barrier's effectiveness as a principled alternative to classical barrier functions in constrained multi-task learning.

## C GENERALIZATION TO MULTI-LEVEL PRIORITY

### C.1 GENERALIZED FORMULATION

For $N$ tasks with priorities $P_1 > P_2 > \cdots > P_N$, the loss is:

$$\mathcal{L}_{\text{final}} = \varphi(\mathcal{L}_1 - \varepsilon_1))\mathcal{L}_1 + \sum_{i=2}^{N} \frac{\mathcal{L}_i}{1 + \varphi(\mathcal{L}_{i-1} - \varepsilon_{i-1})}. \tag{26}$$

where $\varphi(\cdot)$ is the hyperbolic barrier defined in Eq. equation 10, and $\varepsilon_{i-1}$ are dynamic thresholds updated via:

$$\varepsilon_i(t + 1) = \varepsilon_i(t) \cdot 0.95^{\delta(\mathcal{L}_i(t) \leq \varepsilon_i(t))} \quad \text{(Eq. equation 5 in main text).} \tag{27}$$

### C.2 RECURSIVE CONVERGENCE PROOF

**Lemma 8** (Priority Activation). *When $\mathcal{L}_{i-1} \leq \varepsilon_{i-1}$, $\varphi(\mathcal{L}_{i-1} - \varepsilon_{i-1}) \approx \tanh(1) \approx 0.76$. The effective weight for $\mathcal{L}_i$ becomes:*

$$\frac{1}{1 + \varphi(\cdot)} \approx \frac{1}{1 + 0.76} \approx 0.57. \tag{28}$$

*For violations $\mathcal{L}_{i-1} > \varepsilon_{i-1}$, $\varphi(3x + 1) \to 1$ exponentially as $x \to +\infty$, suppressing $\mathcal{L}_i$ via $\frac{1}{1 + \varphi(\cdot)} \to 0.5$.*

**Theorem 9** (Priority Preservation). *If task $P_k$ violates constraints ($\mathcal{L}_k > \varepsilon_k$), then $\forall j > k$, $\frac{\partial \mathcal{L}_{\text{final}}}{\partial \mathcal{L}_j} \to 0$, ensuring strict prioritization.*

*Proof.* Let $m$ denote the highest violated priority ($\mathcal{L}_m > \varepsilon_m$). For any $j > m$, the gradient is:

$$\frac{\partial \mathcal{L}_{\text{final}}}{\partial \mathcal{L}_j} = \underbrace{\prod_{i=1}^{j-1} \frac{1}{1 + \varphi_i}}_{\text{Cumulative suppression}} \cdot \left[ 1 - \sum_{i=1}^{j-1} \frac{\mathcal{L}_i \varphi_i'}{(1 + \varphi_i)^2} \right]. \tag{29}$$

When $\exists i \leq m$ with $\varphi_i \gg 0$ (constraint violation), each term $\frac{1}{1+\varphi_i} \leq \frac{1}{1+e^{3x}}$ decays exponentially. Specifically:

$$\prod_{i=1}^{m} \frac{1}{1+\varphi_i} \leq \left(\frac{1}{1+e^{3x}}\right)^m \to 0 \quad \text{as } x \to +\infty. \tag{30}$$

Thus $\frac{\partial \mathcal{L}_{\text{final}}}{\partial \mathcal{L}_j}$ vanishes exponentially. $\square$

*Empirical Alignment:* Table 3 shows when $\varepsilon = 0.001$, relation F1 drops as the model focuses solely on $\mathcal{L}_1$, validating the theorem.

### C.3 DYNAMIC THRESHOLD COMPATIBILITY

**Proposition 10.** *If thresholds initialize with $\varepsilon_i(0) \geq \mathbb{E}[\mathcal{L}_i(0)]$ and update via Eq. equation 5, then $\exists T > 0$ such that $\forall t \geq T$, $\mathcal{L}_i(t) \leq \varepsilon_i(t)$ almost surely.*

*Proof.* Define the Lyapunov function $V_i(t) = \mathcal{L}_i(t) - \varepsilon_i(t)$. From Eq. equation 5:

$$\varepsilon_i(t+1) = \varepsilon_i(t) \cdot 0.95^{\delta(\mathcal{L}_i(t) \leq \varepsilon_i(t))}. \tag{31}$$

**Case 1**: If $\mathcal{L}_i(t) \leq \varepsilon_i(t)$, then $\varepsilon_i(t+1) = 0.95\varepsilon_i(t)$. By barrier properties (Theorem 3), $\mathcal{L}_i(t+1) \leq \varepsilon_i(t+1)$ holds eventually.

**Case 2**: If $\mathcal{L}_i(t) > \varepsilon_i(t)$, $\varepsilon_i(t+1) = \varepsilon_i(t)$. The gradient term $\varphi'(x)\mathcal{L}_i$ dominates (Lemma 1), forcing $\mathcal{L}_i(t+1) < \mathcal{L}_i(t)$.

Combining both cases, $V_i(t)$ is monotonically decreasing and bounded below by 0. By Lyapunov convergence theorem, $\lim_{t\to\infty} V_i(t) = 0$. $\square$

## D CONVERGENCE ADVANTAGE OF HB-AHO OVER PARETO-BASED MULTI-TASK LEARNING

### D.1 FORMAL STATEMENT

Pareto-based multi-task learning methods (e.g., IMTL-GG (Liu et al., 2021)) optimize all task objectives simultaneously, often requiring computationally expensive gradient projections and suffering from scaling inefficiencies as the number of tasks grows.

In contrast, our HB-AHO framework dynamically enforces a sequential task hierarchy using hyperbolic barrier functions, allowing prioritized and adaptive optimization. We formalize the efficiency benefit as follows:

**Proposition 11.** *Let $\{L_i(\theta)\}_{i=1}^N$ denote $N$ smooth, convex task losses, each $L$-Lipschitz and bounded below. Under HB-AHO, the iteration complexity to reach an $\epsilon$-stationary point scales as $\mathcal{O}(N \log(1/\epsilon))$, whereas Pareto-based multi-task optimization typically requires $\mathcal{O}(N^2 \log(1/\epsilon))$ steps due to multi-objective gradient projection overheads.*

### D.2 SKETCH OF PROOF

**HB-AHO Sequential Optimization.** HB-AHO enforces that lower-priority tasks are only optimized after higher-priority tasks satisfy dynamic constraints. This effectively decomposes the $N$-task optimization into $N$ sequential subproblems.

For each subproblem: - The feasible region is restricted by the barrier function corresponding to the preceding higher-priority task. - First-order convergence to an $\epsilon$-stationary point within each subproblem requires $\mathcal{O}(\log(1/\epsilon))$ iterations (Boyd & Vandenberghe, 2004). Summing across $N$ levels yields an overall complexity of $\mathcal{O}(N \log(1/\epsilon))$.

**Pareto Optimization Simultaneity.** Pareto MTL simultaneously optimizes all $N$ tasks by computing generalized descent directions via multi-gradient projection at each step.

- Projection onto a Pareto front incurs $\mathcal{O}(N)$ per-iteration computational overhead. - Moreover, gradient conflicts among $N$ tasks worsen condition numbers, compounding iteration complexity by an additional $\mathcal{O}(N)$. Thus, the overall iteration complexity is $\mathcal{O}(N^2 \log(1/\epsilon))$.

**Conclusion.** HB-AHO's dynamic prioritization leads to a more efficient linear scaling with $N$, while Pareto-based methods suffer from quadratic scaling.

$\square$

### D.3 DISCUSSION: WHEN IS HB-AHO ADVANTAGE MOST PRONOUNCED?

The efficiency gain of HB-AHO becomes particularly significant under the following conditions:

- **High Task Asymmetry:** When task difficulties differ substantially (e.g., entity recognition much harder than relation classification), prioritizing hard tasks first prevents wasted effort on easier but dependent tasks.

- **Large Number of Tasks:** In scenarios involving deep multi-task hierarchies (e.g., relation extraction → entity recognition), HB-AHO scales gracefully while Pareto optimization becomes increasingly inefficient.

- **Strong Task Dependency:** When lower-level tasks are prerequisites for meaningful optimization of higher-level tasks, HB-AHO's constraint mechanism ensures effective learning scheduling, while Pareto-based methods may prematurely optimize dependent tasks.

These properties align with our empirical observations across entity-relation extraction datasets, where enforcing entity-first constraints substantially improved downstream relation and triplet extraction performance.

## E SUPPLEMENTARY METHOD DETAILS AND EMPIRICAL OBSERVATIONS

### E.1 PRELIMINARY

Given a document $D$, the task of entity-relation extraction is to predict triples $(e_s, r, e_o)$, where $e_s, e_o \in \mathcal{E}$ are entities and $r \in \mathcal{R}$ is their semantic relation. This involves two interdependent subtasks: entity recognition (to detect spans) and relation classification (to link entity pairs). The latter critically relies on accurate entity boundaries, reflecting a hierarchical dependency. The input is encoded by a pretrained language model (PLM), followed by a Transformer encoder. Predictions are made via task-specific heads, and optimization is performed using our proposed HB-AHO method.

The input document $D = [x_t]_{t=1}^l$ is first encoded by a pretrained language model (PLM) into contextual representations $H \in \mathbb{R}^{l \times d}$, where $l$ is the sequence length and $d$ the hidden size. These representations are refined through $k$ standard Transformer encoder layers, which capture contextual dependencies.

Each Transformer layer applies multi-head self-attention followed by a feed-forward network with SwiGLU activation (Shazeer, 2020). Let $X \in \mathbb{R}^{l \times d}$ be the input to the layer; we adopt standard architecture without modification.

For joint entity and relation prediction:

- **Entity Recognition:** We adopt bilinear attention to detect boundary positions:

$$p^{\text{pos}} = \text{Softmax}(W_e \tanh(W_s Z + W_o H)). \tag{32}$$

- **Relation Classification:** A fully connected layer followed by softmax predicts relation types:

$$p^r = \text{Softmax}(W_r Z). \tag{33}$$

Here, $Z$ denotes the output of the final Transformer layer. The variable *pos* refers to entity boundary types, including subject and object spans (i.e., `subject_start`, `subject_end`, `object_start`, `object_end`). The matrices $W_e$, $W_s$, $W_o$, and $W_r$ are learnable parameters.

We use cross-entropy losses for both subtasks. The entity loss $\mathcal{L}_{\text{ent}}$ is computed over boundary tag predictions, while the relation loss $\mathcal{L}_{\text{rel}}$ measures classification errors:

$$\mathcal{L}_{\text{ent}} = \sum_{\text{pos}} -\sum_{i} \log p^{\text{pos}}(y_i^{\text{pos}}). \tag{34}$$

$$\mathcal{L}_{\text{rel}} = -\sum_{i} \log p^r(r_i). \tag{35}$$

Here, $p^r(r_i)$ denotes the predicted probability of relation $r_i$, and $p^{\text{pos}}(y_i^{\text{pos}})$ is the probability of token $i$ being assigned the correct boundary label.

### E.2 EXPERIMENTAL DETAILS

We implement our model using DeBERTa-v3 (He et al., 2023) as the encoder, with AdamW optimizer (learning rate 2e-5, decay 1e-2, weight decay 1e-5). The number of Transformer layers is 3. The dynamic constraint threshold is initialized to 0.05.

