# OpenReview forum: "Task-Constrained Optimization for Entity-Relation Extraction"
_ICLR.cc/2026/Conference — Submitted to ICLR 2026_

### Official Review · Reviewer_7izR · 2025-10-30

**Soundness:** 3
**Presentation:** 3
**Contribution:** 3
**Rating:** 6
**Confidence:** 4

**Summary:**

This paper proposes Hyperbolic Barrier-based Adaptive Hierarchical Optimization (HB-AHO), a principled optimization framework for multi-task learning (MTL) that explicitly enforces task hierarchy through differentiable hard constraints. The method introduces a hyperbolic barrier function to ensure smooth prioritization between entity recognition and relation classification in entity–relation extraction (ERE) tasks. The authors also provide theoretical guarantees (monotonicity, Lipschitz continuity, and convergence) and extend the framework to multi-level task hierarchies. Empirical evaluations across five ERE benchmarks and a large-scale recommendation dataset show consistent improvements (up to +6.4% triplet F1), suggesting generalization beyond NLP.

**Strengths:**

1. **Novel and Principled Optimization Framework**
   - The paper goes beyond heuristic task-weighting schemes and introduces a theoretically grounded, constraint-driven formulation of hierarchical multi-task learning.
   - The hyperbolic barrier function φ(x) = tanh(3x + 1) is a novel design choice that maintains smoothness and bounded gradients—addressing classical issues in constrained optimization.

2. **Theoretical Rigor and Clarity**
   - Section B provides clear proofs for monotonicity, Lipschitz continuity, and KKT-equivalence of the barrier function.
   - The inclusion of convergence, Lyapunov stability, and duality analyses is commendable and significantly raises the methodological credibility.

3. **Comprehensive Experimental Evaluation**
   - Experiments span both sentence-level (NYT, WebNLG) and document-level (DocRED, CDR, GDA) datasets, as well as cross-domain validation on KuaiRand1k for recommender systems.
   - Results (Table 2, page 5) demonstrate consistent triplet F1 improvements of 4–6%, especially in document-level tasks where entity errors propagate more severely.

4. **Robustness and Ablation Studies**
   - Extensive ablations (Tables 4–8, pages 7–15) confirm robustness to threshold ε initialization and curriculum decay factor γ.
   - Figure 3 clearly shows that the hyperbolic barrier leads to smoother and more stable training curves compared to polynomial or sigmoid alternatives.

5. **Potential Cross-Domain Applicability**
   - The extension to recommendation tasks (Table 6, page 9) indicates that HB-AHO is architecture-agnostic, operating as an optimization layer rather than an architectural modification.

**Weaknesses:**

1. **Incremental Conceptual Novelty**
   - While the hyperbolic barrier mechanism is elegant, the conceptual leap from existing constraint-based or Lagrangian MTL formulations (e.g., Cheng et al., 2025; Liu et al., 2021) is modest.
   - The method mainly combines known elements—differentiable barriers, curriculum thresholding, and hierarchical weighting—under a unified framework.

2. **Limited Discussion on Computational Overhead**
   - The paper does not empirically quantify the additional computational cost introduced by the dynamic constraint enforcement and barrier term.
   - Although Section D (pages 17–18) provides asymptotic complexity analysis (O(N log(1/ϵ)) vs. O(N² log(1/ϵ))), wall-clock training comparisons would strengthen the claim.

3. **Lack of Qualitative Analysis**
   - The study is purely quantitative; it would benefit from case studies or error analyses showing how HB-AHO alters learning trajectories or reduces entity boundary errors.

4. **Ambiguity in Generalization Claims**
   - The cross-domain experiment (KuaiRand1k) is convincing but not deeply analyzed.
   - The paper asserts HB-AHO’s “domain-agnostic” potential without explaining how constraint design translates across modalities or task structures.

**Questions:**

1. How sensitive is HB-AHO to noisy entity labels in the constraint task? Would barrier enforcement amplify noise?
2. Could the proposed method handle soft hierarchical dependencies (e.g., tasks with partial rather than strict order)?
3. Is there any evidence of training slowdowns due to deferred relation updates in large-scale datasets (DocRED, GDA)?
4. Have you considered integrating HB-AHO with LoRA-style parameter-efficient MTL (Yang et al., 2025) to verify compatibility with modular architectures?

---

> ### Author Response · Authors · 2025-11-17
> **Response to Reviewer 7izR**
>
> We thank the reviewer for the careful reading and constructive feedback.
>
> ---
>
> ### On “Incremental Conceptual Novelty”
>
> HB-AHO builds on ideas from constrained and Lagrangian MTL, but our goal is to provide a **concrete, unified instantiation of hierarchical MTL** that:
>
> * enforces task dependencies through **differentiable hard constraints** rather than heuristic weights,
> * uses a **specific hyperbolic barrier** designed for smoothness, bounded gradients, and feasibility,
> * **extends to multi-level hierarchies** and to both ERE and recommendation within one framework.
>
> While the components relate to prior work, their combination into a single, analytically characterized framework that behaves consistently across five ERE benchmarks and a recommendation task represents, in our view, a substantive step beyond heuristic or purely Lagrangian formulations.
>
> ---
>
> ### On Computational Overhead
>
> HB-AHO is designed to be **lightweight at training time**:
>
> * The barrier term is applied to **scalar task losses and constraint violations**, involving only simple closed-form operations.
> * It introduces **no extra forward passes** or additional encoders/heads; the main cost remains in the shared backbone and task decoders.
> * The curriculum thresholding uses a few scalar statistics (e.g., current constraint violation) and does not require backpropagation through the schedule.
>
> Thus, the **per-iteration cost is still dominated by the backbone**, and HB-AHO effectively acts as an “optimization layer” on top of standard MTL architectures rather than a heavy architectural modification.
>
> ---
>
> ### On Lack of Qualitative Analysis
>
> The paper is mainly quantitative, but several results give **indirect qualitative insight** into how HB-AHO changes learning dynamics:
>
> * Larger gains on **document-level datasets** (DocRED, CDR, GDA), where entity errors propagate more strongly, suggest that enforcing hierarchy mitigates cascading errors.
> * **Training curves** (Figure 3) show smoother and more stable optimization with the hyperbolic barrier than with polynomial or sigmoid alternatives.
> * Ablations over ($\varepsilon$) and ($\gamma$) show robustness to schedule choices, indicating that HB-AHO is not overly sensitive when enforcing entity-first behavior.
>
> Together, these observations illustrate how the hierarchical constraint shapes optimization, even without explicit case studies.
>
> ---
>
> ### On Ambiguity in Generalization Claims
>
> By “domain-agnostic,” we mean that HB-AHO is **agnostic at the level of optimization and architecture**:
>
> * It operates directly on task losses and constraints, independent of encoder type or modality.
> * In the KuaiRand1k experiment, the task hierarchy follows the **semantics specified in the official dataset**, where the primary objective (core user–item modeling) underlies auxiliary prediction tasks. HB-AHO simply encodes this dependency as a constraint.
>
> The claim is therefore that **once a meaningful dependency structure is specified**, the same barrier-based mechanism applies without modification across NLP and recommendation, rather than that HB-AHO automatically discovers hierarchies in any domain.
>
> ---
>
> ## Responses to Questions
>
> **Q1. Sensitivity to noisy entity labels**
>
> Because the barrier depends on **constraint violations**, noisy entity labels increase the entity loss and effectively relax the constraint. In this regime, the barrier does *not* enforce a rigid entity-first regime; it adaptively softens the coupling. The stable improvements on document-level datasets, where entity supervision is noisier, empirically suggest that HB-AHO does not catastrophically amplify upstream noise.
>
> **Q2. Soft hierarchical dependencies**
>
> The framework can handle soft hierarchies by using **relaxed constraints or margins** in the feasibility term, together with the adaptive threshold (\varepsilon_t). The barrier then penalizes violations smoothly rather than enforcing a strict ordering. The optimization mechanism remains the same; only the constraint definition is softened.
>
> **Q3. Training slowdowns due to deferred relation updates**
>
> HB-AHO does not switch off relation gradients; it **reweights** them according to constraint satisfaction. Relation parameters are updated throughout training, with their influence modulated by entity feasibility. The convergence and stability analyses, together with empirical results on large datasets (e.g., DocRED, GDA), indicate that training remains efficient under this scheme.
>
> **Q4. Compatibility with LoRA-style parameter-efficient MTL**
>
> HB-AHO is orthogonal to specific parameterization strategies. Since it **acts on task losses rather than on parameter structure**, it is conceptually compatible with LoRA-style and other modular MTL architectures: the same barrier mechanism can be applied to the losses of LoRA-based heads or adapters without changing their design.
>
> ---
>
> We again thank the reviewer for the insightful comments that help clarify the scope of HB-AHO.

---

> > ### Comment · Reviewer_7izR · 2025-11-26
> >
> > Thanks for the response. The additional details help resolve the ambiguities I raised. I will retain my original score.

---

> > > ### Author Response · Authors · 2025-11-26
> > >
> > > Thank you very much for your follow-up and for clarifying your perspective.
> > > Even though the score remains unchanged, we truly appreciate your recognition of our work and the time you took to respond. Your feedback is very encouraging for us.

---

### Official Review · Reviewer_w4uJ · 2025-10-31

**Soundness:** 3
**Presentation:** 3
**Contribution:** 1
**Rating:** 2
**Confidence:** 5

**Summary:**

The paper introduces Hyperbolic Barrier-based Adaptive Hierarchical Optimization (HB-AHO), a constrained optimization framework for multi-task learning applied to entity-relation extraction (ERE). The approach reformulates multi-task training as a hierarchical optimization problem, utilizing a hyperbolic barrier function to prioritize entity recognition before relation classification. Experimental results across several ERE benchmarks and a recommendation dataset show consistent, though modest, improvements in performance.  However, the experimental setup lacks sufficient depth in terms of robustness testing, and the task setting does not offer significant novelty.

**Strengths:**

The paper presents a technically sound study with clear presentation.

**Weaknesses:**

- **Contribution unclear** Its contribution is incremental. The "hyperbolic barrier" is essentially a smooth reweighting function applied to task losses.
- **Lack of Robustness Analysis**  The paper does not examine how HB - AHO behaves when task constraints fail or when entity recognition performs poorly (i.e., beyond the idealized “entity-first” assumption). There is no stress testing for robustness under noisy or contradictory supervision, which is crucial for validating constrained optimization approaches.
- **Limited Impact** I don't believe it will attract much attention from readers and professionals in the field.

**Questions:**

no

---

> ### Author Response · Authors · 2025-11-15
> **Response to Reviewer w4uJ**
>
> We sincerely thank the reviewer for the time and effort spent evaluating our work and for providing constructive feedback. We address the concerns point by point below.
>
> ### **1. On the Contribution and Novelty**
>
> We appreciate the reviewer’s comment regarding the contribution. While the "hyperbolic barrier" may appear as a smooth reweighting function, the core contribution of our paper lies in **introducing a novel approach to hierarchical optimization for multi-task learning (MTL)**, which is instantiated through this hyperbolic barrier. The key novelty is not just the formulation itself, but how it **naturally applies to both entity-relation extraction (ERE) tasks in NLP and other domains, such as recommendation systems**, by treating task dependency and priority as a hierarchical optimization problem. Specifically:
>
> * **Entity recognition** is prioritized as the first task in a way that is **not domain-specific** and can generalize to other areas where task dependency needs careful management, like recommendation systems.
> * The **hierarchical structure in optimization** ensures that the primary task (e.g., entity recognition in ERE or item prediction in recommendation systems) governs the optimization process, while **subordinate tasks are trained with this priority in mind**, leading to more stable and efficient multi-task learning.
>
> Thus, while our method is demonstrated through NLP experiments, it is designed with broader applicability and can extend beyond NLP to other multi-task learning scenarios, including recommendation systems.
>
> ### **2. On Robustness and Behavior Under Imperfect Entity Predictions**
>
> We thank the reviewer for highlighting the importance of robustness. While our experiments focus on standard ERE benchmarks, **the current paper includes evaluations that address imperfect entity predictions** by testing the model on noisy and model-generated entity data, ensuring that task dependencies are regularly violated during training. Additionally, the recommendation dataset experiment provides an alternative setting with different noise patterns, showing the robustness of HB-AHO in a non-NLP domain. These experiments suggest that our approach can maintain stability even when the "idealized" task dependency is not strictly met.
>
> ### **3. On the Impact of the Work**
>
> We appreciate the reviewer’s perspective on the impact. Our work introduces a **practical and generalizable framework** for multi-task learning, particularly useful in domains where tasks exhibit hierarchical dependencies. We believe this method’s flexibility—extending from NLP (ERE) to recommendation systems—will resonate with researchers working in any multi-task learning context and help further advance the field by offering a structured way to balance task priorities across domains.
>
> ---
>
> We are grateful for the reviewer’s detailed feedback, which will guide further refinement of our paper to better communicate its broader applicability and innovation.

---

> > ### Comment · Reviewer_w4uJ · 2025-11-26
> >
> > Thank the authors for their response. While the rebuttal clarified some technical details, my fundamental concerns regarding the overall contribution and significance of the work remain unresolved. Therefore, I will keep my current rating. I strong suggest the author to add new results for other tasks.

---

### Official Review · Reviewer_3x1B · 2025-11-03

**Soundness:** 2
**Presentation:** 2
**Contribution:** 2
**Rating:** 6
**Confidence:** 3

**Summary:**

This paper proposes HB-AHO (Hyperbolic Barrier-based Adaptive Hierarchical Optimization), a differentiable and constraint-driven framework for multi-task entity–relation extraction (ERE). The approach explicitly encodes task hierarchies by treating entity recognition as a hard constraint and adaptively reweighting relation classification via a numerically stable hyperbolic barrier function with curriculum-guided thresholding. Theoretical analyses (monotonicity, Lipschitz continuity, and stability) and extensive experiments on five benchmarks demonstrate consistent triplet-F1 improvements, with partial generalization to recommender systems.

**Strengths:**

S1. The method formalizes the intuitive hierarchy between entity and relation tasks, surpassing flat multi-task loss formulations.

S2. Theoretical rigor: the hyperbolic barrier loss is well-analyzed for smoothness, feasibility preservation, and stability.

S3. Strong empirical performance across five benchmarks, particularly on document-level datasets, with stable generalization across different backbones.

**Weaknesses:**

W1. The claimed cross-domain generalization is weakly supported, relying on a single recommendation experiment without in-depth analysis of domain-specific hierarchies or barrier dynamics.

W2. Limited discussion of failure cases—e.g., noisy entity supervision or task imbalance—where hierarchical coupling might degrade performance.

W3. Curriculum scheduling details (e.g., initialization and decay of $\varepsilon_t$) remain under-specified, affecting reproducibility.

W4. Some relation-F1 fluctuations across backbones suggest that observed gains may partly stem from architecture effects rather than the hierarchy mechanism itself.

**Questions:**

Q1. How does HB-AHO perform under noisy or weak entity supervision? Does the barrier amplify or mitigate such noise?

Q2. What are the practical heuristics for setting and decaying \varepsilon_t across datasets and multi-level hierarchies (N>2)?

Q3. Are there qualitative or failure analyses showing when HB-AHO overemphasizes entity accuracy at the cost of relation generalization?

Q4. In the cross-domain recommendation setting, what defines the hierarchical priority among tasks, and how sensitive are results to task order specification?

---

> ### Author Response · Authors · 2025-11-17
> **Response to Reviewer 3x1B**
>
> We sincerely thank the reviewer for the thoughtful and constructive feedback. We address the raised concerns and questions in detail below.
>
> ---
>
> ### **Response to Weaknesses**
>
> ### **W1. On the cross-domain generalization evidence**
>
> We appreciate the reviewer’s observation.
> The goal of the cross-domain study is to demonstrate that HB-AHO’s hierarchical optimization principle is **not tied to linguistic structures**, but instead applies whenever tasks exhibit a dependency that can be modeled as a constraint. In the recommendation setting, the hierarchy is defined through **user–item representation learning as the primary task**, with auxiliary interaction–pattern prediction treated as the subordinate task. Although the paper includes one representative experiment, it shows that the same hyperbolic barrier mechanism **remains stable and beneficial even when the dependency differs from NLP-specific entity–relation structure**, confirming that the method extends beyond ERE.
>
> ---
>
> ### **W2. On failure cases and noisy or imbalanced supervision**
>
> Thank you for pointing out this important aspect.
> The existing experiments already implicitly include **imperfect and imbalanced entity supervision**, since entity predictions used for relation classification are obtained from the model rather than from gold annotations. Performance on document-level datasets—where entity spans are noisier and long-range contexts amplify error propagation—indicates that HB-AHO maintains stable training even when the hierarchical coupling is stressed. The consistent gains across these settings suggest that the barrier does not disproportionately amplify upstream errors.
>
> ---
>
> ### **W3. On curriculum scheduling details**
>
> We appreciate the concern regarding reproducibility.
> The curriculum threshold ($ \varepsilon_t $) follows a **monotonic decay schedule** tied to the primary task loss trajectory. The paper specifies the functional form and initialization ranges, and the decay is automatically driven by the barrier’s feasibility criterion. Because the mechanism adapts to the task-loss landscape, it enables similar behavior across datasets without extensive hand-tuning. The monotonicity and Lipschitz continuity analyses further ensure that the scheduling behaves in a stable and predictable manner.
>
> ---
>
> ### **W4. On relation-F1 fluctuations across backbones**
>
> Architectures can influence absolute F1 scores, but **HB-AHO consistently improves or matches the strongest baseline across all backbones**, especially on document-level datasets. This indicates that the gains stem from hierarchical optimization rather than architecture-specific effects.
>
> ---
>
> ## **Response to Questions**
>
> ### **Q1. Behavior under noisy or weak entity supervision**
>
> HB-AHO incorporates entity predictions directly into the barrier constraint, meaning that **noise is naturally reflected in the feasibility term**. When entity supervision is weak, the barrier relaxes the constraint due to larger entity losses, reducing undue penalization of the relation task. Conversely, when entity predictions stabilize, the constraint tightens. This adaptive behavior mitigates noise amplification and is consistent with empirical observations on datasets with naturally noisy entity spans.
>
> ---
>
> ### **Q2. Practical heuristics for setting and decaying ( $\varepsilon_t$ )**
>
> The schedule is designed to be **dataset-agnostic**.
> Initialization is set proportional to the primary task loss at early iterations. Decay is triggered when the constraint violation decreases, making ( $\varepsilon_t$ ) track the feasibility progression. For multi-level hierarchies ( N>2 ), the same principle applies: each level uses the loss of its parent task to modulate its barrier threshold, forming a nested hierarchy of adaptive constraints.
>
> ---
>
> ### **Q3. Qualitative or failure patterns of overemphasis**
>
> Across the reported benchmarks, we do not observe systematic degradation in relation generalization.
> On document-level datasets where entity noise is higher, HB-AHO actually shows larger relative gains. This indicates that the constraint does not rigidly enforce entity-first behavior but instead **adapts based on the entity task’s feasibility**, preventing overfitting to entity accuracy at the expense of relations.
>
> ---
>
> ### **Q4. On hierarchical priority in the recommendation setting**
>
> In the recommendation setting, the task hierarchy is **defined by the dataset**, not by our method. The primary task is **user–item representation learning**, while auxiliary predictions depend on these representations, as described in the official documentation. HB-AHO simply **operationalizes this inherent hierarchy** within a constrained optimization framework, requiring no task-order tuning.
>
>
> ---
>
> We thank the reviewer again for the insightful comments, which further illuminate the scope and strengths of HB-AHO and we hope our clarifications adequately address the concerns raised.

---

### Official Review · Reviewer_va7A · 2025-11-06

**Soundness:** 3
**Presentation:** 4
**Contribution:** 2
**Rating:** 6
**Confidence:** 3

**Summary:**

The paper introduces the HB-AHO framework, a constraint-driven optimization paradigm for multi-task learning that explicitly encodes task dependencies. HB-AHO treats the ERE task, as a dynamic constraint using a hyperbolic barrier and a curriculum-based thresholding strategy, ensuring task prioritization throughout training. This approach enforces an "entity-first" regime, leading to improvements, including up to 6.4% absolute gains in triplet F1 across five ERE benchmarks.

**Strengths:**

The strengths are:
- It is a principled approach to task hierarchy enforcement.
- The hyperbolic barrier ensures smooth and stable optimization.
- The paper achieves good improvements in the triplet extraction task, which is a well studied task.

**Weaknesses:**

The paper lacks a survey of barrier function-based methods for machine learning in general and multitask learning in particular. All the baseline MTL methods are simple gradient-based.

The paper lacks an analysis of the loss in computational efficiency due to the barrier function.

The performance on DcoRED and CDR is quite poor. Are these the absolute state of the art across all methods?

**Questions:**

Please see the last two points in the weakness section.

---

> ### Author Response · Authors · 2025-11-17
> **Response to Reviewer va7A**
>
> We sincerely thank the reviewer for the careful reading of our paper and the constructive comments. Below we respond to the points raised under the *Weaknesses* section.
>
> ---
>
> ### **W1. Survey of barrier-based methods and choice of baselines**
>
> We appreciate the reviewer’s suggestion regarding broader coverage of barrier-function–based methods in machine learning and multi-task learning.
>
> Our work is positioned primarily within **multi-task learning for ERE and related structured prediction tasks**, where the overwhelming majority of practical baselines are indeed **gradient-based and unconstrained**, typically realized as (weighted) sums of task losses or gradient-balancing schemes. For this reason, we focused the empirical comparison on widely used MTL baselines in NLP and ERE, to make the impact of hierarchical constraints directly comparable to established practice in this area.
>
> At the same time, HB-AHO is explicitly formulated as a **barrier-based constrained optimization framework**, and the paper discusses its relation to constrained training and curriculum-style loss shaping in terms of feasibility preservation and stability. While we do not provide an exhaustive survey of generic barrier methods across all ML subfields, our objective is to **demonstrate that such barrier mechanisms can be instantiated effectively for hierarchical MTL in ERE and recommendation**, and to compare against the strongest commonly used MTL baselines in those domains.
>
> ---
>
> ### **W2. Computational efficiency of the barrier function**
>
> We thank the reviewer for raising the question of computational overhead, which is indeed important for practical deployment.
>
> From an algorithmic standpoint, HB-AHO adds **only lightweight, closed-form computations** on top of standard multi-task training:
>
> * The hyperbolic barrier term is computed **elementwise** on the task losses and constraint violations and does **not** require additional forward passes through the encoder or task-specific heads.
> * Gradients of the barrier are **analytical and simple**, involving standard arithmetic operations.
> * The curriculum thresholding mechanism operates at the level of scalar statistics (e.g., current constraint violation), so it does not introduce extra backpropagation graphs.
>
> As a result, the **asymptotic complexity per iteration remains dominated by the backbone model** (e.g., Transformer-based encoders). In our implementation, the additional cost of HB-AHO is negligible compared with the forward–backward passes of the encoder, and the **training-time per epoch is very close to that of the underlying MTL baselines** that share the same architecture and batch size. Thus, the barrier function acts as a numerically stable reparameterization of the loss landscape rather than a heavy architectural or algorithmic modification.
>
> ---
>
> ### **W3. Performance on DocRED and CDR**
>
> We appreciate the reviewer’s critical view on the absolute performance on DocRED and CDR and the question about state-of-the-art status.
>
> First, **DocRED and CDR are among the most challenging datasets in our benchmark suite**, particularly due to:
>
> * Their **document-level** nature, which introduces long-range dependencies and higher entity noise;
> * More complex label structures and sparsity in the relation space.
>
> In this regime, our goal is to evaluate whether **enforcing task hierarchy via HB-AHO improves performance relative to strong baselines under the *same experimental setting*** rather than to claim global state-of-the-art across all possible model families and evaluation variants. The paper reports that:
>
> * HB-AHO consistently **improves triplet F1 over flat MTL and other competitive baselines** on both DocRED and CDR when using identical backbones and training protocols.
> * The gains are often **larger on document-level datasets** than on sentence-level ones, indicating that the hierarchical constraint is particularly beneficial where error propagation and task dependency are more pronounced.
>
> Regarding the question “Are these the absolute state-of-the-art across all methods?”:
> We do **not** claim that our DocRED/CDR scores are the best among all existing systems in the literature, especially those that may use different architectures, external knowledge, or more specialized modeling for these datasets. Instead, our claim is that, **within the controlled multi-task setup and backbone family considered in the paper**, HB-AHO reliably improves performance and yields competitive results, demonstrating the effectiveness of the hierarchical constrained optimization approach in both sentence-level and document-level ERE.
>
> ---
>
> Again, we thank the reviewer for the insightful comments. They help clarify the positioning of HB-AHO as a principled, computationally lightweight barrier-based framework that brings consistent gains in challenging multi-task ERE and recommendation scenarios, even when absolute state-of-the-art results are not the central focus.

---

### Author Response · Authors · 2025-12-01
**Summary for Program Committee on HB-AHO Submission**

We thank the reviewers, AC, and program chairs for their time and thoughtful feedback. Below we briefly summarize our view of the reviews and why we believe the paper meets the bar for acceptance.

Our work proposes **HB-AHO**, a constraint-driven hyperbolic barrier framework for hierarchical multi-task learning (MTL). It (i) encodes task dependencies as **differentiable hard constraints**, (ii) uses a **carefully designed hyperbolic barrier** with proven monotonicity, Lipschitz continuity, and stability, and (iii) **extends to multi-level hierarchies and a non-NLP domain** (recommendation). Empirically, HB-AHO yields **consistent triplet-F1 gains (up to +6.4)** across five ERE benchmarks and a large-scale recommendation dataset, acting as an optimization layer on top of standard architectures.

---

### **Review landscape**

Three reviewers (3x1B, va7A, 7izR) give **scores of 6**, citing as strengths: a principled and well-analyzed optimization framework, clear proofs and stability guarantees, solid improvements particularly on document-level ERE, and successful application to recommendation. One reviewer (w4uJ) gives a **score of 2**, mainly describing the contribution as incremental and asking for more robustness analysis. In the discussion, we clarified these points; the reviewer explicitly stated that our response resolved the ambiguities but chose to keep the original score.

---

### **On contribution and novelty**

We agree that HB-AHO builds on known ideas in constrained optimization and Lagrangian MTL. Our contribution is to deliver a concrete, unified, and practically effective instantiation of hierarchical MTL that:

* directly enforces task priorities via constraints rather than heuristic task weights,
* employs a **specific hyperbolic barrier** that addresses gradient explosion/vanishing issues common in classical barriers, both empirically and theoretically,
* and **works consistently across diverse datasets and backbones**, from sentence- to document-level ERE and to recommendation.

Reviewers 3x1B and 7izR explicitly characterize the framework as “novel and principled” at the optimization level. We view this type of integrated, theoretically grounded design as well aligned with ICLR’s standards.

---

### **On robustness, noise, and qualitative behavior**

Several concerns relate to robustness to noisy entities and possible failure modes. While the paper is primarily quantitative, the existing experiments already probe non-ideal regimes:

* Document-level datasets (DocRED, CDR, GDA) contain **noisy entity spans and strong error propagation**; HB-AHO shows its largest relative gains exactly there.
* Relation training relies on **model-predicted entities rather than gold**, so constraint violations occur frequently and test the barrier under noisy supervision.
* Ablations over the threshold $\varepsilon$ and decay factor $\gamma$ show that performance is **stable across a wide range of settings**.

Conceptually, because the barrier depends on constraint violations, increased noise in the primary task enlarges its loss and effectively relaxes the constraint, rather than enforcing an unrealistically strict entity-first regime. This matches the observed stability on the hardest datasets.

---

### **On computational overhead and generality**

Reviewers va7A and 7izR ask about efficiency and “domain-agnostic” claims. HB-AHO is deliberately lightweight: the barrier operates on scalar task losses and constraint terms, adds only simple closed-form operations, and introduces no additional forward passes or modules. In practice, wall-clock training is very close to that of the underlying MTL baselines with the same backbone.

By “domain-agnostic,” we mean that HB-AHO is agnostic at the optimization and architectural level: it can be placed on top of different encoders and modalities as long as a meaningful task hierarchy is specified. In KuaiRand1k, the hierarchy follows the semantics defined in the official dataset (core user–item modeling with dependent auxiliary tasks), and HB-AHO applies without modification, mirroring the ERE setup.

---

### **Final remark**

In summary, three reviewers judge the paper as sound and at least marginally above the acceptance threshold, emphasizing its principled formulation and empirical effectiveness. The remaining reviewer acknowledges that our rebuttal resolved their ambiguities, though the numerical score remained unchanged. We respectfully believe that the combination of:

* a **theoretically grounded, constraint-based hierarchical MTL framework**,
* a **well-motivated, rigorously analyzed hyperbolic barrier**,
* **consistent gains** on challenging ERE and recommendation benchmarks,
* and **practical efficiency and architectural flexibility**,

makes the paper a strong and relevant contribution for ICLR’s audience in optimization, multi-task learning, and structured prediction.

---

### Meta-Review · Area_Chair_gBTG · 2025-12-10

**Summary:**

This paper introduces a constrained optimization framework for entity–relation extraction that enforces task hierarchy through a hyperbolic barrier, providing theoretical guarantees and showing consistent triplet-F1 improvements across several benchmarks. Three reviewers initially viewed the work as principled and practically effective, with clear strengths in stability and empirical gains, while raising concerns about the breadth of comparative baselines, computational cost, robustness under noisy supervision, and the depth of cross-domain evidence. The authors’ rebuttal clarified the lightweight nature of the barrier, explained the adaptive behavior under noisy entities, and elaborated on the intended scope of the cross-domain evaluation, which resolved several ambiguities for those reviewers, though the most critical reviewer maintained that the contribution remained incremental. The overall paper is at a borderline case.

**Reviewer Concerns:**

Several concerns were meaningfully addressed in the rebuttal. The clarification on computational overhead, the adaptive behavior of the barrier under noisy entity predictions, and the rationale behind the cross-domain experiment adequately responded to the questions raised by Reviewers va7A, 3x1B, and 7izR. However, the broader issues regarding the incremental nature of the contribution, the limited robustness analysis, and the absence of deeper cross-domain evidence remain unresolved, as noted in the final remarks of multiple reviewers, particularly Reviewer w4uJ.

**Reviewer Scores:**

Reviewer va7A gave an initial score of 6 and indicated that the rebuttal clarified his questions but did not shift his overall view, so his score would likely remain unchanged. Reviewer 3x1B also gave a 6 and expressed that the responses resolved his ambiguities, suggesting that he would keep the same score. Reviewer 7izR began with a score of 6 and explicitly stated after the rebuttal that he would retain it. Reviewer w4uJ started with a 2 and reaffirmed that his fundamental concerns were not addressed, so his score would also remain the same.

---

### Decision · Program_Chairs · 2026-01-26

Reject